# Nanoformulations-Based Metronomic Chemotherapy: Mechanism, Challenges, Recent Advances, and Future Perspectives

**DOI:** 10.3390/pharmaceutics15041192

**Published:** 2023-04-08

**Authors:** Vijay Kumar Panthi, Kamal Dua, Sachin Kumar Singh, Gaurav Gupta, Philip M. Hansbro, Keshav Raj Paudel

**Affiliations:** 1Department of Pharmacy, College of Pharmacy and Natural Medicine Research Institute, Mokpo National University, Jeonnam 58554, Republic of Korea; 2Discipline of Pharmacy, Graduate School of Health, University of Technology Sydney, Sydney, NSW 2007, Australia; 3Faculty of Health, Australian Research Centre in Complementary & Integrative Medicine, University of Technology Sydney, Ultimo, NSW 2007, Australia; 4School of Pharmaceutical Sciences, Lovely Professional University, Phagwara 144411, India; 5School of Pharmacy, Suresh Gyan Vihar University, Mahal Road, Jagatpura, Jaipur 302017, India; 6Centre for Inflammation, Faculty of Science, School of Life Sciences, Centenary Institute and University of Technology Sydney, Sydney, NSW 2050, Australia

**Keywords:** nanoformulations, cancers, metronomic chemotherapy, clinical study

## Abstract

Cancer-related death is a significant health and economic burden worldwide, and some conventional chemotherapy is associated with limited effectiveness in completely curing various cancers, severe adverse effects, and destruction of healthy cells. To overcome the complications associated with conventional treatment, metronomic chemotherapy (MCT) is extensively suggested. In this review, we aim to highlight the importance of MCT over conventional chemotherapeutic approach with emphasis on nanoformulations-based MCT, their mechanism, challenges, recent advances, and future perspectives. Nanoformulations-based MCT revealed remarkable antitumor activity in both preclinical and clinical settings. For example, the metronomic scheduling of oxaliplatin-loaded nanoemulsion and polyethylene glycol-coated stealth nanoparticles incorporating paclitaxel were proven very effective in tumor-bearing mice and rats, respectively. Additionally, several clinical studies have demonstrated the benefit of MCT with acceptable tolerance. Moreover, metronomic might be a promising treatment strategy for improving cancer care in low- and middle-income nations. However, an appropriate alternative to a metronomic regimen for an individual ailment, suitable combinational delivery and scheduling, and predictive biomarkers are certain parts that remain unanswered. Further clinical-based comparative research studies are mandatory to be performed before entailing this treatment modality in clinical practice as alternative maintenance therapy or in place of transferring to therapeutic management.

## 1. Introduction

Cancer is a leading cause of death and a key hurdle to enhancing life expectancy globally [1,2]. Cancer prevalence is continuously growing in high-income countries and, particularly, in low and middle-income nations [3]. However, the rate of cancer survival is ameliorating in developed countries. The factors associated with improvements depend on the types of cancer, access to evidence-based screening, early diagnosis, and amenities in treatment and supportive care [4]. A huge number of cancer patients will continue to be detected each year for decades; a predicted 14 million patients in 2012 globally, with a 50% estimated to rise to 21.6 million patients annually by 2030. Before curing cancer, various treatment modalities were approached, including chemotherapy, surgery, radiation therapy, targeted therapy, or a combination. The term “chemotherapy” was introduced by Paul Ehrlich, a German chemist who evaluated the application of drugs to cure infectious disorders. Moreover, he was also the initial researcher to investigate a range of chemicals to examine their potential activity against ailments using animal models. Additionally, historical evidence reveals that arsenicals application commenced in the 1900s, while in the 1960s, surgery and radiotherapy was the cornerstone of cancer management. Furthermore, after the appearance of cancer recurrence and micrometastases owing to radiation therapy and surgery, combinational chemotherapy started achieving beneficial effects [5]. But recently, several therapeutic pathways associated with the progression of cancer and their targeted mechanisms have remarkably ameliorated illness. The combined approaches, including multiple targeted modalities or conventional chemotherapeutics (platinum compounds and taxanes) observed to have a synergistic impact. In addition, new treatment strategies, such as biological molecules, drugs, and immune-based therapies, are applied for curing. Chemotherapy is assumed to be the most beneficial and broadly applied modality in curing cancers as a single delivery or combined with radiotherapy [6,7,8]. However, despite traditional chemotherapy being beneficial to some extent, the vital demerits of chemotherapy are its high-dose requirements, poor bioavailability, adverse effects, marginal therapeutic indices, non-specific targeting, and development of multiple drug resistance (MDR) [9]. To overcome the drawbacks associated with conventional chemotherapy, a new therapeutic strategy called “metronomic chemotherapy” (MCT) has been proposed. This treatment approach deals with the chronic, equally spaced delivery of low strengths of several chemotherapeutic drugs without expanded rest durations. Additionally, the novelty of this therapeutic strategy exists not only in its effectiveness against tumors with extremely minimal toxicity but also in a cell target switch, now focusing on tumor-endothelial cells. The skills and expertise obtained in the preclinical area of MCT, plus the enhanced experience acquired in the clinical practice, will aid in introducing a change in the design of therapeutic guidelines against cancer [10,11]. On the other hand, the emergence of nanotechnology has had a crucial influence on medical treatment in the previous two decades. Compared with traditional cytotoxic agents, nano-drug delivery systems have revealed the potential to circumvent a few of these problems by ameliorating therapeutic effectiveness while preventing toxicity in healthy cells owing to characteristics including maximum selective deposition in tumors through improved permeation and accumulation impact and active cellular uptake [12,13]. Among several nanoformulations drug delivery systems, polymeric nanoparticles, liposomes, and micelles have exhibited significant promising clinical impacts. These days, various nanoparticle-mediated cytotoxic drugs are clinically permitted, and several are in different phases of preclinical and clinical progression. Although nanoscale drug carriers facilitate tremendous benefits as drug delivery systems, their poor bioavailability, lack of biodegradation, insufficient bio-distribution, instability in the circulation, and potential toxicity raise concerns over their safety, particularly for long-lasting delivery. Chemoresistance, considered one of the frequently appearing cases in cancer treatment, is a process in which cancer cells that are originally suppressed by an antineoplastic drug appear resistant to the specific drug. Due to this, new drug carriers with better targeting potentiality are required for cancer prevention, the minimization of adverse effects, and pain management related to cancer treatment [9,14]. Additionally, enzyme-responsive nanoparticles have been assumed as one of the most encouraging and fashionable stimulus-responsive nanoparticles. Bioenzymes that aid in catalyzing the reactions within the living systems reveal a massive promise for cancer treatment. At the same time, they are integrated with nanoparticles to ameliorate their deposition into tumor portions. Nanomedicines can administer harmful bioenzymes into tumor tissues to directly cause their demise in cancer therapy. Firstly, modifications in the expression of specified enzymes, including phosphatases, proteases, and glycosidases, could be observed in inflammatory areas, which can be repressed to obtain a targeted collection of drugs at the preferred biological site through enzyme-derived drug release [15,16,17]. Further, in case of economic concerns, the majority of drugs utilized in the metronomic regimen have generic equivalents, are inexpensive, and are available in oral form, which prevents expensive hospital stays and intravenous injections. The minimal expenditure related to MCT represents an opportunity for applying this treatment choice, particularly in developing nations. It poses a difficulty for the commencement of massive trials financed by industry [18,19]. Even in the developed nations, increasing expenditure on cancer therapy is a crucial public and political issue. In low- and middle-income nations (LMINs), where most new cancer cases and cancer mortality occur, the approach to various modern treatment modalities is beyond imagination. Although standard health and medicine care is better in high-income nations, access to health and medicine care by LMINs is limited, owing to their costs, toxicity, and the complicated infrastructure and technology required. As some research proposed previously, metronomic might be a promising optional modality for the augmentation of cancer care in LMINs. This therapeutic approach seems even more optimistic in a maintenance setting where the administration is required for the longer term and can be more beneficial due to the acceptable toxicity profile and affordable expenditure [20,21]. In this review, we discuss metronomic chemotherapeutic approach-consisted nanoscale drug delivery systems for treating several cancers in preclinical and clinical settings with special emphasis on recent advances and challenges associated with MCT, followed by a discussion about the future directions of MCT.

### 1.1. Methodology

The search for relevant scientific literature was conducted in the search engines such as PubMed, google scholar, Scopus, and Web of Science database using the aforementioned keywords (in abstract). The literature includes original articles, reviews, and book chapters published in English from January 2000 to March 2023. A few critically important papers published before January 2000 were also included.

### 1.2. Conventional Chemotherapy and Its Limitation

Conventional chemotherapy uses chemical compounds as a single or multi-drug to treat disease, notably cancer, until maximum cytotoxicity is achieved by administering the maximum tolerable dose (MTD). It can be used in conjunction with other cancer treatments like radiation and surgery [22]. The prevalent conventional cancer treatment approaches include tumor removal with surgery, followed by radiotherapy and X-rays, and/or chemotherapy. Surgery is exceedingly efficacious during the beginning phase of disease progression among these treatment strategies. Similarly, radiotherapy can impair healthy cells, tissues, and organs. To achieve 100% eradication of cancer cells with tolerable toxicity and manageable side effects, cytotoxic drugs are delivered intravenously at the MTD [22,23,24]. Despite decreasing morbidity and mortality rates by chemotherapy, almost all anticancer agents destroy healthy cells, particularly speedily-dividing and proliferating cells. Therefore, determining methods to enhance efficacy and minimize side effects would outstandingly ameliorate the potentiality of cancer therapy [25].

Cancer stem cells (CSCs) are frequently responsible for tumor chemoresistance [26]. These tumor-initiating cells can self-renew and account for a small percentage of the heterogeneous tumor. In addition, other drawbacks associated with the traditional chemotherapeutic approach are difficulty in dosage preference, inadequate precision, rapid drug metabolism, and especially detrimental side effects. Notably, metastatic return after chemotherapy has been postulated because of therapeutic resistance, especially in CSCs, as their apoptosis evasion permits the tumor to regress following treatment. The process that regulates drug availability, epithelial-mesenchymal transitions (EMT), and oncogenic signaling pathways involve numerous mechanisms that allow cancer cells to resist the lethal effects of medicines [27]. Additionally, cancer cells have developed the ability to amplify genes and modify gene expressions through mutations in the coding genes for apoptosis-inducing proteins, resulting in medication resistance. Further, changes in the function or structure of proteins involved in drug transport impact the amount of drug reaching cancer cells, exposing the tumor cells to decreased drug concentrations and conferring resistance [28]. As a result, a new therapeutic method known as MCT was developed, in which 1/10th–1/3rd of the MTD is often given without long drug-free periods [29].

## 2. Metronomic Chemotherapy

The MCT deals with the frequent administration of anticancer agents relatively at low doses in a repeated manner and without allowing a more extended period of the drug-free state. Previously, it was assumed that MCT target angiogenesis to manage cancer progression; however, in recent times, alternative mechanisms have been identified discovering MCT as part of a multi-targeted treatment strategy. Furthermore, the scientific principle for MCT is that traditional anticancer drugs focus enlargement of the vascular endothelial cell. Still, the effect against angiogenesis cannot be prolonged due to the endothelial cells getting an opportunity to retrieve while breaking the treatments, and this may be overwhelmed by therapy at low doses in a repeated manner. Additionally, Hanahan et al. introduced the term ‘metronomic’, acquired from the word “metronome”, a melodic tool that generates continuous metric ticks as a stable, constant aural pulse. MCT is the repeated delivery of antineoplastic drugs at minimal doses with no extended drug-free interval. Thus, it achieves a sustained and lower drug concentration in the blood without prominent harmful side effects. There are some main properties of MCT, including repeated drug administration at low strengths without any breaks, utilizing a biologically-optimized dose over the MTD without applying hematopoietic growth factors, and drug preference via oral route [30,31].

### 2.1. Mechanism of Action

MCT is a multi-targeted therapy that exhibits both straight and incidental impacts on tumor cells and their microenvironment. Additionally, it can impede angiogenesis in the tumor, activate the immune response against cancer and enhance tumor dormancy (Figure 1) [30].

#### 2.1.1. Angiogenesis Inhibition

MCT demonstrates its anti-cancer property primarily by hampering tumor angiogenesis. The procedure of metronomic drug delivery has been documented to augment the significant in vitro potentiality of some antineoplastic drugs against angiogenesis, including cyclophosphamide (CPM) and taxanes. Additionally, the anti-angiogenic mechanism action of MCT has been revealed in vivo that encompasses apoptosis proliferation of stimulated endothelial cells is specifically hindered, as well as inhibition of endothelial cell migration. Moreover, other associated mechanisms are enhanced thrombospondin -1 (innermost barrier of angiogenesis) expression and prolonged decline in bone marrow-acquired endothelial progenitor cell levels and viability [33].

#### 2.1.2. Triggering of Immunity

Both intrinsic and acquired immune systems have a pivotal impact on the progression and suppression of cancer. In the case of the immune system, the most prevalent adverse reactions of chemotherapy are lymphopenia and neutropenia. However, considerable research has shown and assessed that few anticancer drugs, such as CPM, taxanes, and anthracyclines, reveal immune-triggering characteristics [34]. In this regard, the influence on modulatory T cells is moderately compatible with metronomic treatments. Moreover, T cells are CD^+^ CD25^+^ Foxp3^+^ lymphocytes which can impede antigen-specific immune response in two different methods, such as cytokine and cell contact-dependent manner. Therefore, T cells can retard immune response on the tumor by inhibiting the action of both tumor-specific (CD8^+^ cytotoxic T lymphocytes and CD4^+^ T helper cells) and tumor-unspecific effector cells such as natural killer (NK) and NK T cells. Further, regarding the various human cancers, T cells have been obtained in elevated proportions corresponding to tumor development and inadequate treatment response [35]. Thus, deterioration of T cells activity, either by specified interruption or reduction, is a procedure to increase the immune response of antigens that are associated with the tumor. Additionally, several experimental and clinical studies have proved the impact of low-dose CPM on T cells. It decreases the proportion of T cells, hampers its function, and accelerates the proliferation of lymphocyte and memory T cells [30,36].

#### 2.1.3. Initiation of Tumor Dormancy

Tumor dormancy exists due to cell-cycle interruption or from an influential balanced condition in which the initiation of apoptosis balances cell enlargement. Tumor dormancy can be found during the beginning stage of cancer development and after the accomplishment of anticancer therapy during the remission stage, where tumors can recommence their progression from persisting residual ailment [37]. Angiogenesis inhibition, programmed cell death of cancerous cells, and immune surveillance are three prime techniques by which MCT promotes tumor dormancy [38].

### 2.2. Chemotherapeutic Difference: Conventional and Metronomic

As per the traditional chemotherapy regimens, cytotoxic drugs are given in cycles near or at the maximum tolerated dose. They substitute with a considerable drug-free break to permit the patient to recover from severe drug reactions. This approach effectively impedes the disease process in an extensive number of both pediatric and adult patients but corresponds to some difficulties. Additionally, although improvement is usually achieved in the initial stage, recurrence is a prevalent issue in the case of metastatic and endangered cancers [30,39]. On the contrary, MCT means repeated and continuous utilization of conventional chemotherapeutics at low doses, an advancing option to traditional chemotherapy [40]. Moreover, its appropriate toxicity profile, lower cost, and convenient application are vital over conventional treatments [41]. The overall difference between metronomic and conventional chemotherapy is revealed in Table 1.

### 2.3. Nanoformulations-Based MCT in Pre-Clinical Setting

The application of nanotechnology to ameliorate therapeutics is not new; indeed, there has been a significant enhancement in nanotechnology research as the advantages become more evident. This field has uninterruptedly broadened into radiotherapy, chemotherapy, prognosis, and imaging, demonstrating the potential to extend and advance patient care [43]. In addition, nanomedicines are utilized as optional therapeutics for antineoplastic agents. For the management of cancer, owing to the lesser particle size in nanometers (nm), specified site targeting can be accomplished by applying nanomedicines, enhancing their bioavailability, and deliberating minimal harmful side effects. Moreover, the utilization of a minute quantity of drugs aid in cost savings [44]. Additionally, nanotechnology is extensively used in fabricating such agents as they are in nm sizes, believed to be one of the pioneer cutoff values for manufacturing products in nanotechnology. Currently, anticancer nanomedicines that have been approved are mainly liposomal preparations and drug conjugates (polymer, protein, and/or antibody) aimed at augmenting the pharmacokinetics and pharmacodynamics of the free drug and applying passive targeting. To date, various cytotoxic drugs such as oxaliplatin (OXA), pemetrexed (PMX), etoposide (ETP), and docetaxel (DTX), were incorporated in different nanocarriers, including nanoemulsion, micelles, niosomes, etc. to examine their anticancer efficacy in an experimental setting [6,43].

Like other experimental treatments, MCT assembled its groundwork with numerous experimental research, commencing with the innovatory work in the laboratories of Folkman and Kerbel. Judah Folkman initiated that enhancement of tumor angiogenesis was necessary for malignant advancement, and a later angiogenesis-promoting factor, vascular epidermal growth factor (VEGF), was separated from tumors. Subsequently, Folkman suggested that impeding the synthesis of VEGF could deprive the tumor, which is defined as a dormant tumor. The impacts of a metronomic regimen consisting of a single drug have exhibited moderate success when combined with other cytotoxic drugs like CPM. In tumor-bearing mice (tumors obtained from breast cell lines), CPM delivered regularly in drinking water with an antiangiogenic agent demonstrated remarkable efficacy against tumors and could be beneficially useful to chronic therapy [45,46]. Various preclinical studies highlighted the efficacious significance of MCT and led the path to their clinical assessment. In current decades, considerable research has been carried out to shed light on the therapeutic effectiveness of MCT and its mechanism of action. Originally, it was assumed that a metronomic regimen acted solely on active proliferating cells, specifically on the endothelial cells of the tumor vasculature. The development of new vessels in the enlarging tumor is needed for proliferation, supplements, and oxygen. A study performed by Miller et al. during the early 20th century proposed two different mechanisms of anticancer drugs. One is chemotherapeutic agents can act directly against the cancer cells, which are actively enlarging, and another approach is indirectly acting over the newly formed vessels [47]. With the influence of this recommendation, Browder and team advised a modality to encourage the anti-angiogenic impacts of chemotherapy: an uninterrupted delivery of antineoplastic agents at doses below MTD without considerable gaps between phases. They exhibited that a regular metronomic regimen of CPM at a low dose is more efficacious than the standard schedule in cultured breast cells, which have attained drug resistance [48]. Further, a study done by Klement and colleagues has shown that long-term delivery of low-dose vinblastine in combination with antibodies consisting of anti-vascular endothelial growth factor-2 (VEGFR-2) appeared in tumor suppression [49]. Recently, Maharjan and team evaluated the immune-modulating efficacy of novel oral MCT consisting of OXA and PMX for colon cancer in a mouse model. In this study, MCT did not reveal considerable lymphotoxicity, while MTD was associated with systemic immunosuppression. Further, MCT encouraged the proportion of functional T cells in the tumor. In combination with anti-PD-1 (anti-programmed cell death protein-1), MCT demonstrated suppression of tumor volume by 97.85 ± 84.88% of the control group, resulting in a 95% complete response [50]. Additionally, the same author and team also performed another metronomic treatment of oral PMX-incorporated colloidal dispersions to increase antitumor immune response in cancer-bearing mice. In this study, metronomic oral administration of PMX showed higher immunity against tumors in addition to the elevating ratio of infiltrating lymphocytes and hindering T-cell functions [51]. In addition, research performed by Choi et al. examined the tumor-regulating immunity of OXA-loaded nanoemulsion (OXA-NE) via metronomic scheduling in tumor-bearing mice. In this study, tumor volume was impeded by 78.3 ± 9.67% compared to the control group. Further, OXA-NE exhibited immunomodulatory efficacy, including elevation of tumor antigen uptake and boosting the function of immune effector cells in tumor tissue [52]. Furthermore, a study by Jha and colleagues prepared DTX-loaded NE and examined antitumor efficacy in tumor-bearing mice via metronomic dosing. In this study, the optimized formulation demonstrated a remarkably higher tumor-impeding rate in the treatment group as compared to the control. Moreover, oral absorption was ameliorated, resulting in the enhanced chemotherapeutic effect of DTX via MCT [53]. Additionally, a study by Pangeni et al. examined oral absorption and anticancer efficacy using OXA-loaded solid formulation and the efficacy of oral metronomic scheduling on colorectal cancer in rats and monkeys. The particle size of the optimized powder formulation was obtained within the nanoscale range (133 ± 1.47 nm). In both animal models, the findings revealed considerable enhancement in oral absorption in addition to the extensive impedance of tumor growth via metronomic dosing [54]. Furthermore, research done by Cai and colleagues assessed the anti-angiogenic and anti-tumor efficacy of novel zoledronic acid-loaded liposomes via metronomic scheduling in a mouse model. Metronomic therapy has demonstrated an extensive hampering effect on tumor progression. Further, this therapy impaired tumor-associated macrophages by hindering CD206 antibodies in tumor tissues [55]. In addition, Jyoti and colleagues performed in vitro evaluation of liposomal topotecan-activating metronomic therapy combined with radiation in tumor-endothelial spheroids. The findings suggested that liposomal incorporation of topotecan elevates the effect by shielding it from systemic clearance, permitting higher uptake, and enhanced tissue exposure in tumors [56]. Further, another study by Cai and the team examined the anti-angiogenic and antitumor efficacy of triptolide in nude mice via a combined approach of metronomic administration and target delivery. In this study, the triptolide-loaded liposome exhibited the better anti-tumor effect of target metronomic delivery compared to liposome administration at MTD, evidenced by a remarkable decline in tumor volume, vessel density, and the volume of circulating endothelial progenitor cells in serum [57]. In addition, research performed by Fei et al. developed polyethylene glycol (PEG)-coated stealth nanoparticles incorporating paclitaxel (PCL) prior to assessing the tumor accumulation and their antitumor activity via metronomic schedule in the Sprague-Dawleyrat model. The findings exhibited that PCL-loaded nanoparticles ameliorated tumor deposition and antitumor effect following a metronomic regimen [58]. Furthermore, Doi and colleagues studied the combined treatment of a metronomic S-1 regimen with OXA consisting of PEG-coated liposomes ameliorates activity against tumors in a colorectal tumor-bearing murine model. The combined oral delivery of metronomic S-1 dosing with an intravenous administration of OXA-loaded liposomal formulation showed considerable antitumor properties without serious overlapping adverse effects in comparison with a metronomic regimen of S-1, free OXA, and OXA liposomal formulation alone [59]. Further, another study by Hoelzer and the team evaluated the tumor-targeting metronomic effect of doxorubicin (DXB) using pH-responsive poly (2-oxazoline)-based nanogels in nude mice via tail vein injection. In this study, poly (2-oxazoline)-based nanogels demonstrated a remarkable inhibition on tumor progression and enhancement in survival duration, whereas pure DXB had no impact on tumor growth [60].

### 2.4. MCT in Clinical Practice

To date, many studies have revealed that most MCT-based clinical trials were acceptably tolerated. Additionally, this treatment strategy is considered an alternative to cost-benefit treatment as assessed in several clinical trials. The severe toxic effects are either almost not found or very rare. In general, adverse effects were vomiting, mild to moderate leucopenia, lymphopenia, neutropenia, and anemia, in addition to minor-grade fatigue. Furthermore, MCT provides an outstanding clinical advancement, including an upgrade in the quality of life with negligible toxicity. However, to augment the MCT use in clinical practice, we require a well-performed phase III study showing the considerable significance of metronomic dosing. Moreover, well-evaluated randomized phase II studies would also lead to convincing evidence to promote the application of low-dose chemotherapy. The number of people to be studied should be correspondent and should associate with patients representing the real world [20,61]. Thus far, it is still difficult to find clinical trials that have a randomized setup and are statistically significant. In this review, we highlighted the clinical study of MCT in recent years, mainly on breast cancer, lung cancer, colorectal cancer, prostate cancer, gastroesophageal cancer, and pancreatic cancer.

#### 2.4.1. Breast Cancer

Breast cancer is possibly one of the most adequately studied tumors for MCT. The therapy is generally targeted at patients suffering from metastatic disorder to ameliorate the quality of life and mitigate ailment symptoms. MCT offers the best substitute for traditional chemotherapy with minimal side effects and efficacy. The most repeatedly examined cytotoxic drugs in metronomic trials were CPM, methotrexate (MTX), vinorelbine (VRB), and capecitabine (CPB). The most adequately tested anticancer agents as a monotherapy are CPB and VRB [31]. In 2010, Taguchi et al. evaluated the efficacy of CPB at a low dose as first-line treatment in 33 metastatic breast cancer patients. This drug was regularly administered as per the following schedule: 825 mg/m^2^ twice daily for 21 out of 28 days. The overall survival (OS) and median progression-free survival (PFS) were 24.8 and 6.9 months, respectively. This study exhibited that CPB was better tolerated and efficacious in this schedule [62]. Fedele and colleagues examined the effect of frequent CPB by administering 1500 mg once daily in 60 pretreated metastatic breast cancer patients. This study revealed the median time to progression (TTP), median OS, and clinical benefit rate (CBR) was 7, 17 months, and 62%, respectively. Moreover, VRB was assessed at 70 mg/m^2^ thrice a week in 34 elderly patients, followed by one week break. This schedule of MCT was adequately tolerated (grade 3 neutropenia 6%) with an OS, and median FPS were 15.9 and 7.7 months, respectively [63]. In addition, De Iuliis et al. assessed VRB in 32 patients at a dose of 30 mg via oral at every alternate day. This study showed that the safety profile was optimal (no grade 3 or 4 events), and CBR was 50%. Similarly, combinational oral administration of CPM and MTX was delivered at a low dose to 63 patients, revealing an overall CBR and overall objective response rate (ORR) of 32% and 19%, respectively [64].

#### 2.4.2. Non-Small Cell Lung Carcinoma

Lung cancer is also among the most prevalent cancers globally, constituting almost 20% of cancer mortality [65]. Although there is an advancement in both targeted- and immune therapies, many lung cancer patients still get chemotherapy during their oncological ailment [66,67]. Traditional chemotherapy may not be the appropriate alternative for debilitated or geriatric patients owing to its toxicity. This problem can be overcome by MCT, an efficacious and better-tolerated schedule for this subtype sufferer. The most studied antineoplastic agent in non-small cell lung carcinoma (NSCLC) is VRB, used as monotherapy via oral delivery, which has been examined in various phase II studies. The most followed schedule was monotherapy, thrice a week, specifically for patients considered unsuitable for platinum-based treatments or already receiving several therapies [31]. The data provided by multicenter international retrospective evaluation on 270 NSCLC patients who received metronomic VRB via oral route at the dose of 50 mg, 40 mg, and 30 mg three times a week as first, second, or subsequent line, respectively, demonstrated activity of disease stabilization for a longer period with an overall disease control rate (DCR) and response rate (RR) of 61.9% and 17.8%, respectively [68]. In addition, another study evaluated 30 mg oral VRB three times a week for chemotherapy-receiving patients, showing 4 months of median PFS and 26% and 7 months of ORR and median OR, respectively [69]. Similarly, Goern and colleagues used a weekly schedule of 25 mg/m^2^ DTX and 50 mg trofosfamide once daily in 62 patients with IV NSCLC. In this study, median OS, overall response, and PFS were 9.6 months, 19%, and 2.9 months, respectively [70]. In another study, 100 mg oral ETP was given to NSCLC patients. The corresponding figures for stable disease and partial response were 34% and 28%, respectively, in addition to median OS of 9 months and 6 months of TTP [71]. A metronomic schedule of gemcitabine and PCL was examined in combination with bevacizumab with alternate evidence of vascularization. In 39 patients with advanced NSCLC, the median PFS rates were observed at 61% and 21% for 6 and 12 months, respectively, while ORR was 56% and median OS was 25.5 months [72].

#### 2.4.3. Colorectal Cancer

Colorectal cancer is also considered one of the most prevalent cancers and a leading cause of cancer death worldwide [73]. Fluoropyrimidines are the cornerstone of ideal chemotherapy in colorectal cancer. They are accepted as the primary therapy for metastatic colorectal cancer until getting the permission of novel anticancer agents such as OXA and irinotecan and biological drugs including cetuximab, bevacizumab, and panitumumab. Regorafenib and trifluridine/tipiracil (TAS-102) have recently been certified as therapeutic alternatives for profoundly investigated metastatic colorectal cancer. However, these therapeutic approaches might be inappropriate for some geriatric patients and patients treated previously in an enormous manner where disease control is necessary with a high safety profile and better quality of life [31]. Various research showed that MCT is an efficacious and safe approach for this subtype of sufferers. In addition, several types of anticancer drugs were examined in colorectal cancer as MCT. These agents are irinotecan, CPB, and CPM; however, the most appropriate findings were obtained from the oral delivery of CPB. From 1998 onwards, a continually specified daily regimen of CPB has been assessed as a treatment choice in colorectal cancer. A retrospective evaluation carried out on 50 patients administering an uninterrupted constant dosage regimen of irinotecan or fluorouracil (1500 mg or 2000 mg daily) with or without other treatments revealed a minor toxicity profile, and none of the patients treated with a metronomic schedule of CPB experienced any grade of side effects [74]. Moreover, a more current retrospective evaluation of 68 previously treated patients with a metronomic regimen of CPB at a daily dose of 1500 mg exhibited a median OS and disease control rate of 8 months and 26%, respectively [75].

Additionally, a randomized phase III clinical trial exhibited that maintenance therapy and metronomic schedule of CPB plus bevacizumab succeeding six cycles of traditional CPB + bevacizumab + OXA markedly ameliorated PFS in comparison with the monitoring group (11.7 vs. 8.5 months) without damaging quality of life. These findings highlight the effect of metronomic CPB as a therapeutic choice in geriatric and heavily pretreated metastatic cancer; however, its impact as maintenance therapy requires further examination [41,76].

#### 2.4.4. Prostate Cancer

In the previous three decades, several MCT-based treatments have been assessed in advanced-stage prostate cancer (PC) sufferers. However, most research was performed before the commencement of new hormonal or anticancer drugs, including apalutamide, abiraterone acetate, darolutamide, enzalutamide, or cabazitaxel. All the aforementioned antineoplastic agents have been permitted to manage castration-resistance prostate cancer (CSPC) patients contingent on large, well-assessed phase III clinical trials. In contrast, the significance of MCT has been generally examined in small, non-randomized trials or retrospective examinations. Similarly, MCT should not be applied as a standard treatment modality at the beginning phases of metastatic CRPC if extremely active, better-tolerated new hormonal agents are easily accessible. Thus, MCT illustrates a healing alternative in metastatic CRPC who are unsuccessful in all accessible treatments, e.g., in low or middle-income nations (LMINs) [77]. A recent study by Calvani and colleagues exhibited that CPM 50 mg orally once a day as a metronomic regimen with corticosteroids (oral delivery of dexamethasone; 1 mg/day or prednisolone; 10 mg/day) enhanced biochemical responses in 50% of the studied population. In this study, single-agent CPM resulted in a moderately low rate of prostate-specific antigen responses (16%). In addition, corresponding figures for OS and progression-free survival were 8.1 and 4 months, respectively [78]. In another study, CPM 100 mg with etoposide 50 mg (14/21 days) was examined in 20 hormone refractory sufferers where OR was reported at 35% [79].

#### 2.4.5. Gastroesophageal Cancer

Despite having the appropriate accessible therapies, patients with gastroesophageal cancer (GEC) have a poor diagnosis. Patients are exhausted and might be unable to tolerate or progress on standard cytotoxic chemotherapy [80]. There are various causes for the improper diagnosis of GEC patients. This cancer is considered a systemic disorder even at the beginning phase. Research performed by Bobek and the team demonstrated the appearance of circulating tumor cells in approximately 70% of patients with resectable esophageal cancer [81]. Despite having no randomized clinical trials that reveal the impact of MCT in the management of GEC, several case reports and case studies have demonstrated the indication of MCT, particularly in palliative practice [82]. A retrospective study in 47 patients, 25 patients of progressive GEC receiving capecitabine 1500 mg once a day continuously, revealed therapeutic safety with partial response (PR) and stable disease (SD)in 1 and 6 patients, respectively [83]. Besides, capecitabine 1000 mg/day, (days 1–28 continuously, every 5 weeks) was assessed in another prospective study consisting of 45 elderly patients of pretreated GEC. In this research, the median time to progression (mTTP), median OS, and RR were 3.6 months, 7.6 months, and 20.9%, respectively. The toxicity profile was achieved within the acceptable limit with lesser than 10% grade 3 and no grade 4 toxicities [82,84].

#### 2.4.6. Pancreatic Cancer

Most pancreatic tumors originate from the exocrine pancreas, which has increased proportionately in Western nations. Since the past couple of years, most novel selective agents focused against specified cellular targets have become accessible for cancer treatment, leading to remarkable advancements. However, despite achieving such improvements, the prognosis is still disappointing [85]. A weekly low-dose metronomic regimen of nab-paclitaxel (60 mg/m^2^) and oxaliplatin (50 mg/m^2^) in addition to continuous infusion of 5FU (180 mg/m^2^/d) was retrospectively investigated in 65 patients who have advanced pancreatic cancer (PC) patients. This metronomic regimen was proven efficacious and safe in which RR, OS, and DCR were 49%, 19 months, and 81%, respectively [83,86]. Alongside, a 1500 mg once-a-day metronomic regimen of capecitabine was also examined in 22 pretreated patients with pancreato-biliary cancer, in which a greater survival rate was achieved in patients with 33% SD [84].

### 2.5. MCT as Maintenance Therapy

MCT is the standard treatment to accept as a maintenance modality after commencing standard chemotherapy, mainly owing to its uninterrupted schedule and multifunctional potentiality. However, cost-profitable and quality of life are two other pivotal problems that correspond with the chronic use of maintenance therapy. The hypothesis of metronomic maintenance is not novel. The effectiveness of this strategy in acute lymphoblastic leukemia is the foremost and time-assessed demonstration. The impact of MCT on the tumor microenvironment, such as anti-angiogenesis and modulation of the immune system, may be the finest practice in the maintenance approach, while the ailment burden is minimal. The chemo-switch strategy of multi-targeted metronomic is the evidence of conception for this approach. A recent study by Xiangwei and colleagues demonstrated that maintenance therapy promoted extending PFS and OS in NSCLC patients. In this study, maintenance therapy was beneficial and well-tolerable for patients who suffered from advanced NSCLC, particularly adenocarcinomas without distant metastasis, which were pretreated with targeted anticancer drugs, which was an invariable prognostic element for OS. The median OS and PFS were markedly extended in the maintenance therapy group as compared with the non-MT group [87]. In addition, research performed to examine the impact of metronomic VRB in NSCLC as a switch maintenance approach after first-line platinum-based chemotherapy showed an extended PFS compared to best supportive care [88]. Another meta-analysis of 22 clinical trials of 1360 patients highlighted the promising outcomes of MCT in advanced breast cancer patients [89]. Furthermore, various studies with MCT were already done in preliminary breast cancer, specifically in triple-negative sufferers, and as maintenance therapy after adjuvant modality [90,91]. In a recent study by Ji-Bin et al., the one-year maintenance therapy of CPB for preliminary phase triple-negative breast cancer after standard treatment, the comparison was made with routine follow-up, the result was obtained remarkably cost beneficial with optimistic clinical significances and better-enhanced costs [92].

### 2.6. MCT as Targeted Therapy

Indeed, MCT is a multi-targeted therapy that exhibits its impacts on cancer cells and their microenvironment via direct and indirect ways (Table 2). It acts by impeding tumor angiogenesis, activating an immune response against tumor cells, and enhancing tumor dormancy [30]. Targeted therapy is a category of cancer treatment that uses antineoplastic agents to target specified proteins and genes that aid the proliferation, multiplication, and migration of tumor cells. The combinational approach of MCT with targeted therapy can ultimately escalate the effectiveness and precision of maintenance, as revealed in different experimental settings and the clinic. It is worth noting that the combined efficacy of targeted therapies with MCT has been inconsistent and demonstrated primarily unsuccessful. At the same time, a combination was given with anti-angiogenic tyrosine kinase inhibitors (TKIs) compared to monoclonal antibodies. Accordingly, various laboratory and preliminary clinical research have demonstrated that MCT could be beneficially combined with pazopanib, an oral multikinase inhibitor revealing promoting action in pediatric solid tumors and gynecological cancer. In such a case, MCT can be applied as a bridge to fulfill the space among anticancer drugs and biologicals. The possible significance of chronic utilization of targeted angiogenic inhibitors and MCT might be effectively used in maintenance. The objective of MCT is not merely palliative therapy; a neoadjuvant setting is an effective field of examination. A study performed by Dellapasqua and the team demonstrated the findings of CPM in combination with liposomal DXB as a preliminary treatment in locally advanced breast cancer sufferers. The ratio of breast-conserving surgery was obtained at 44.8%. Further, patients showed 62.1% PR without revealing grade 4 toxicity [93].

An immunogenic response was also assessed using CPM, MTX, and an immunogenic vaccine. The ORR achieved was 23.8%. Besides, research carried out by Young and the team evaluated the CPB with DTX once a week to a metronomic dose of taxane therapy to enhance the expression of thymidine phosphorylase with a once-a-day regimen of celecoxib. In this study, the observed CB and median time of disease progression (TTP) were 42% and 3.6 months, respectively [94]. Anti-angiogenic activity of microtubule inhibitors is valuable as a part of metronomic treatment. The oral delivery of VRB has been examined in geriatric patients of metastatic breast cancer, in which ORR reported was 38% [95]. An alternate dose of 100 mg oral ETP was estimated in NSCLC patients; stable disease and PR were 34% and 28%, respectively. Additionally, the corresponding figures for OS and TTP were 9 months and 6 months, respectively [71]. Another metronomic regimen of ETP with lomustine and CPM was administered via oral route in 71 pretreated small cell lung carcinoma (SCLC) patients. In this study, the ORR reported was 38% and severe; however, hematological toxicity was rare [96].

## 3. Recent Advances and Challenges

A metronomic regimen is utilized to provide uninterrupted chemotherapy at minimal doses. The minimal doses have fewer adverse effects and may promote cancer therapy to be rearranged toward the path of chronic disease management [61]. It is true that despite considering a promising approach, MCT as maintenance has yet to be adequately examined in clinical practice. Most of the research is small with single-arm observations. However, a better safety profile and optimistic effects have been revealed by many of these evaluations. There are various continuing clinical trials that are assessing the metronomic influence as maintenance in several combinations, such as breast cancer, ovarian cancer, and colorectal cancer. The findings obtained from these trials will further justify the metronomic application as maintenance therapy [20]. Further, until now, most of the studies related to metronomic treatment did not exhibit remarkably ameliorated OR and/or progression-free survival. Fundamentally, the metronomic regimen was evaluated in a maintenance setup and was concomitantly administered with other agents within normal high doses, whereas no trial was carried out challenging the metronomic regimen and appropriately encouraging care in later therapeutic lines. Thus, there is no solid proof of the better effect of a single regimen of MCT, and firm evidence of this treatment modality is missing until this date. Additionally, there is a requirement for further verification of the effectiveness of this strategy in a clinical setting [61]. There are some studies on the immune impacts of anticancer agents with recommendations that MCT associates with immunological action and activity against angiogenesis. Specifically, there is a curiosity about promoting the death of an immunogenic cell in checkpoint inhibition. Moreover, in the case of repurposed agents, the hypoglycemic agent metformin has been considered extensive research attentiveness, with retrospective examinations revealing survival significance in diabetic patients with cancer. Out of various anticancer mechanisms, there is proof of an antiangiogenic response with apoptosis enhancement in breast cancer cell lines. Initial results advised that metformin treatment diminishes specific numbers of immunosuppressive cells and may reveal antagonist impacts when followed with treatment of a checkpoint inhibitor. On the contrary, until now, the unpublished outcome reports that metronomic treatment may coordinate with checkpoint inhibition, facilitating an opportunity for novel combinational therapies. To conclude, despite having these intermittent concerns, the field of MCT exists progressive and pioneering. In addition, clinical trials are expanding, entailing all ailment practices, not merely the palliative. There is a considerable fascination with the feasibilities of combinational delivery of MCT with repurposed non-cancer agents. There is a modernized curiosity in co-delivery with immunotherapeutic and targeted therapies [20,31].

On the other hand, cost-efficiency and quality of life are two major problems related to the long-term utilization of metronomic treatment as maintenance. The financial influence of maintenance for advanced cancers has been evaluated in various cost-efficient examinations. Apart from this, tolerance and compliance with prolonged therapy are vital issues, but second-line treatment on disease advancement may also accelerate worldwide worsening of quality of life mainly because of ailment-associated symptoms, which can consequently compromise the possibility of second-line therapy. Until now, no clinical trial has exhibited a remarkable amelioration in quality of life with maintenance therapy. However, different research has demonstrated that global quality of life does not degrade, revealing the acceptable toxicity profile of extended delivery of a single anticancer drug or biological agents [20,97]. Undoubtedly, the expenditure on cancer treatment is a serious economic burden and an urgent matter to be solved. The financial difficulty of cancer therapy is due to more costly drugs [98]. A study by Bocci and the team compared the results and healthcare-associated budgets of metronomic dosing with new therapeutic modalities in breast cancer. This study, which is the solely pharmacoeconomic assessment of metronomic treatment, efficiently exhibited the possibility of MCT as more cost beneficial. MCT is more favorable over MTD, with minimal doses, less parenteral administrations, and minor complex proportions and, therefore, a lesser necessity for infrastructure. Considering the tumor and the patient in cancer care, it may be feasible to integrate the administered drugs and dosing regimen; metronomic schedules are more suitable for this as compared to MTD. The comparison of the effects of MTD and MCT is shown in Figure 2. Indeed, this tailoring course requires a response and surrogate marker to supervise the MCT. Although few have been reported, up to now, the absolute prognostic biomarker has not been acknowledged in the literature. This is a shortcoming of metronomic treatment as its treatment actions are not applicable for direction [41,99]. To augment the application of MCT in clinical practice, a well-performed phase III study that detects a scientifically remarkable significance of metronomic regimen is required. The appropriately performed randomized phase II studies would also carry captivating documentation to promote the utilization of low-dose cancer treatment. In addition, the research population should be homogeneous and encompass sufferers representing the real world. Alternatively, it is essential to correlate a single metronomic drug with either current high-dose chemotherapy or the best symptomatic treatment in a randomized setting to examine the effect of a metronomic regimen with rigorous investigation. Therefore, we still need more trials with analytical stability and a randomized foundation. Taken together, although both MCT and drug repurposing suffer from difficulties of an economic impediment in high-income countries, the minimal expenditure related to these therapies is a positive in LMINs. Admittedly, with progressive positive findings, we may observe medical practitioners in high-income economies learn from these circumstances [20,61].

## 4. Future Perspectives

The findings obtained from experimental and clinical research have documented MCT as a novel therapeutic modality to impede specified categories of malignancies. However, optimization of metronomic anticancer treatment is challenging until this date, even after the completion of clinical evaluation for more than a decade. Thus, in the near future, cancer research should focus on recognizing the most appropriate therapeutic agents to utilize for the management of tumors, prior to determining the doses of each drug to be applied single or combined, as well as to interpret the scheduling of drug delivery. Further, new approaches are being established in medical oncology to conjugate MCT with traditional cancer treatment, targeted therapy, and/or radiotherapy. Such modalities can expand numerous opportunities for effective combinations. Meanwhile, a detailed study of pharmacoproteomic and pharmacogenetics on tumor-endothelial cells is now required to examine their vulnerability to MCT and to assess the utmost beneficial drug combination to utilize in the medical [30]. On the other hand, the ever-enhancing expenditure of cancer therapy is a major community and political concern even in the advanced global. In LMINs, where most cancer incidences and mortality occur, the approach to many recent treatment strategies is far from imagination. Although fascinating, the developed nations’ standards of care are beyond accessible for LMINs owing to their expenditure, fatalness, and the complicated framework and technology required. As proposed by previously published studies, metronomic might be an optional optimistic modality for the augmentation of cancer management in LMINs. This therapeutic approach seems highly promising even in a maintenance setting where prolonged delivery is mandatory and can be more beneficial due to its acceptable toxicity profile and affordable price [20]. Moreover, approximately a couple of decades ago, metronomic treatment approaches have been justifiably evaluated for substituting, improving, or appending traditional regimens in several cancers. Until now, most published research is experimental, phase I, and phase II. It is feasible to illustrate that, for the most part, those data are reasonably corresponding to standard regimens but do not hypothesize the clinical results; phase III studies are necessary to initiate the impact of MCT for cancer management. Currently, the more acceptable modality is ongoing examinations and gradual incorporation of metronomic therapy into our latest practice as an alternative to sweeping it aside [20,101].

## 5. Conclusions

These days, the MCT modality is extensively applied as a targeted therapy in both preclinical and clinical settings which promotes researchers to deliver the drug to the targeted tumor site via various nanoscale drug delivery systems. Therefore, several complications associated with conventional chemotherapy were solved. The better toxicity profile, lower cost, and convenient application are vital merits of MCT over conventional treatment. The evidence of MCT for personalized medicine is increasing, commencing with unfit elderly patients and for palliative therapy. As we discussed in this review, with the utilization of the MCT approach, various drug-incorporated nanoformulations revealed remarkably higher antitumor activity compared to the free drug in different stages of clinical trials as well as in preclinical practice. Combining metronomic chemotherapy with specified targeted treatment might ultimately increase the effectiveness and specificity of maintenance, as shown in various experimental models and the clinic. However, the choice of the appropriate metronomic regimen for an individual ailment, suitable combinational delivery and scheduling, predictive biomarkers, and the time arrangement of the organization are certain parts that still exist unanswered. As per the global oncology perspective and achievement of recent optimistic findings, metronomic might be a promising treatment modality for the augmentation of cancer management in LMINs. Additionally, comparisons with the appropriate supportive care are inadequate, and therefore, we urgently wait to introduce the pivotal role of MCT in treating frail patients. Further clinical-based comparative research studies necessary to be carried out before encompassing metronomic regimens in clinical settings as an alternative maintenance treatment or in place of shifting to remedial care.

## Figures and Tables

**Figure 1 pharmaceutics-15-01192-f001:**
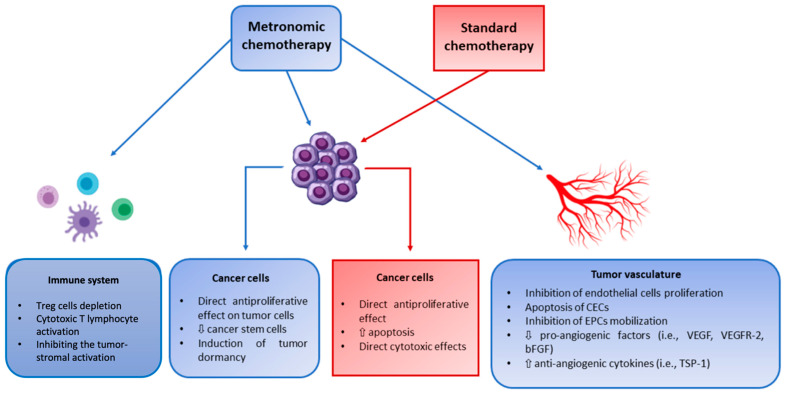
Mechanism of action of metronomic chemotherapy versus standard chemotherapy. bFGF; basic fibroblast growth factor, CEC; circulating endothelial cell, EPC; endothelial progenitor cell, TSP-1; thrombospondin-1, T_reg_; regulatory T cells, VEGF; vascular endothelial growth factor. Reproduced with slight modification and permission from [32].

**Figure 2 pharmaceutics-15-01192-f002:**
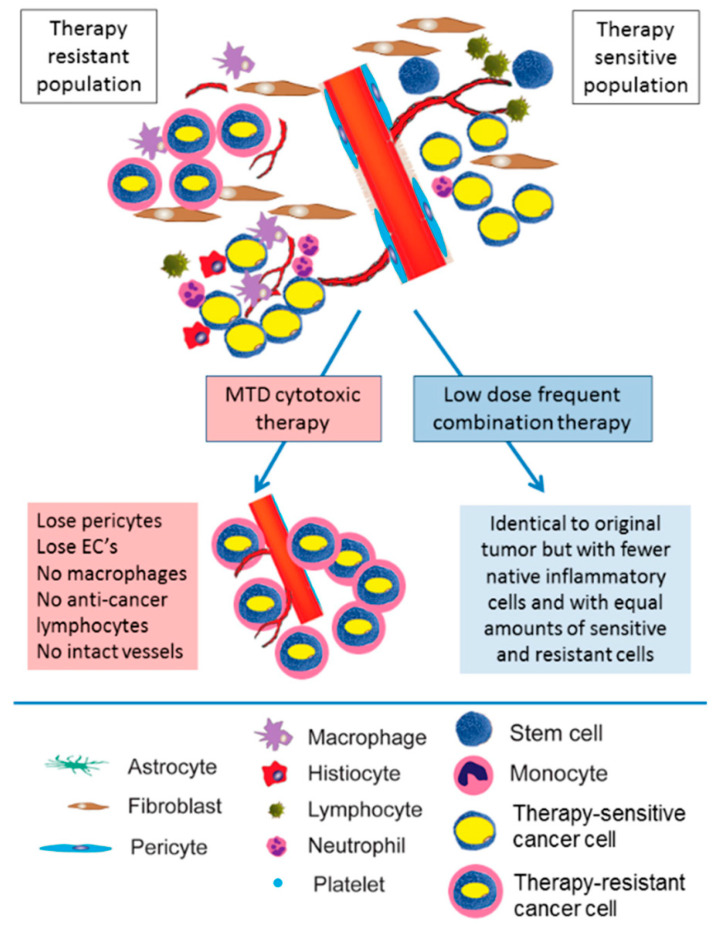
Comparison of the effects of maximally tolerated dose (MTD) and metronomic chemotherapy (MCT). MTD cytotoxic chemotherapy results in the ablation of anti-tumor immunity and the elimination of therapy-sensitive clones. In turn, these result in the selection of therapy-resistant cancer cells. In contrast, low-dose high-frequency (metronomic) chemotherapy targets the tumor stroma, gradually reducing tumor size but mostly maintaining tumor composition, decreasing therapeutic resistance, and maintaining anti-tumor immunity. Figures adapted from [100].

**Table 1 pharmaceutics-15-01192-t001:** Difference between conventional chemotherapy and MCT.

Parameters	Conventional Chemotherapy	MCT	References
Dose	Designed to deliver at MTD	Below than MTD	[38]
Dosing frequency	Specified intervals (weekly, fortnightly, three-weekly)	Daily, every alternate day, weekly	[30]
Plasma concentration	Fluctuated	Sustained	[30,42]
Target cells	Proliferating tumor cells	Endothelial cells (in the progressing vasculature of the tumor)	[33]
Aim of administration	To treat cancer directly by hampering or destroying speedily dividing cancer cells	To achieve cancer control by focusing on angiogenesis	[30]
Toxicity	Toxicity is a main concern due to drugs used as MTD	No remarkable toxic side effects as achievement of sustained low drug level in blood	[38]
Efficacy	More successful in the primary tumor in comparison with metastasis	Advanced cancer	[30]

MTC = metronomic chemotherapy, MTD = maximally tolerable dose.

**Table 2 pharmaceutics-15-01192-t002:** Metronomic chemotherapy for various cancers.

Cancer Types	Metronomic Chemotherapy	Findings	Reference
Breast cancer	Capecitabine: 825 mg/m^2^ twice daily for 21 out of 28 days.	The overall survival and median progression-free survival were 24.8 and 6.9 months, respectively. This study exhibited that CPB was better tolerated and efficacious in this schedule.	[62]
	Vinorelbine: 70 mg/m^2^ thrice a week in 34 elderly patients, followed by one week break.	The regimen was adequately tolerated (grade 3 neutropenia 6%), and overall survival and median progression-free survival were 15.9 and 7.7 months, respectively.	[63]
	Combinational oral administration of cyclophosphamide and methotrexate delivered at a low dose to 63 patients.	The overall clinical benefit rate and overall objective response rate were 32% and 19%, respectively.	[64]
Non-small cell lung cancer	Vinorelbine via oral route at 50 mg, 40 mg, and 30 mg three times a week as a first, second, or subsequent line.	Vinorelbine demonstrated activity of disease stabilization for a longer period with an overall disease control rate and response rate of 61.9% and 17.8%, respectively.	[68]
	Weekly schedule of 25 mg/m^2^ docetaxel and 50 mg trofosfamide once daily in 62 patients of IV NSCLC	The median overall survival, overall response, and progression-free survival were 9.6 months, 19%, and 2.9 months, respectively.	[70]
	Etoposide: 100 mg oral was given to NSCLC patients.	The corresponding figures for stable disease and partial response were 34% and 28%, respectively, in addition to median overall survival of 9 months and six months to progression.	[71]
Colorectal cancer	The uninterrupted constant dosage regimen of irinotecan or fluorouracil (1500 mg or 2000 mg daily) with or without other treatments.	A minor toxicity profile and none of the patients treated with a metronomic schedule of capecitabine experienced any grade of side effects.	[74]
	Metronomic regimen of capecitabine at a daily dose of 1500 mg.	Median OS and disease control rate of 8 months and 26%, respectively.	[75]
Prostate Cancer	Cyclophosphamide 50 mg orally once a day as a metronomic regimen with corticosteroids (oral delivery of dexamethasone; 1 mg/day or prednisolone; 10 mg/day).	Cyclophosphamide resulted in a moderately low rate of prostate-specific antigen responses (16%). Moreover, corresponding figures for overall survival and progression-free survival were 8.1 and 4 months, respectively.	[78]
	In another study, CPM 100 mg with etoposide 50 mg (14/21 days) was examined in 20 hormone refractory sufferers.	The overall response was reported to be 35%.	[79]
Gastro-esophageal cancer	Capecitabine 1500 mg once a day continuously.	They revealed therapeutic safety with partial response and stable disease in 1 and 6 patients, respectively.	[83]
	Capecitabine 100 mg/d, d1-28/35.	Median time to progression, median overall survival, and response rate were 3.6 months, 7.6 months, and 20.9%, respectively. The toxicity profile was achieved within the acceptable limit with lesser than 10% grade 3 and no grade 4 toxicities.	[84]
Pancreas cancer	A weekly low-dose metronomic regimen of nab-paclitaxel (60 mg/m^2^) and oxaliplatin (50 mg/m^2^) in addition to continuous infusion of 5FU (180 mg/m^2^/d).	This metronomic regimen proved to be efficacious and safe in which RR, OS, and DCR were 49%, 19 months, and 81%, respectively.	[86]

NSCLC = non-small cell lung cancer, CPM = Cyclophosphamide, 5FU = % fluoro-uracil, CPB = Capecitabine, RR = response rate, OS = overall survival, DCR = disease control rate.

## Data Availability

No new data were created or analyzed in this study. Data sharing is not applicable to this article.

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
