# Peer review of "Nanoformulations-Based Metronomic Chemotherapy: Mechanism, Challenges, Recent Advances, and Future Perspectives"

_pharmaceutics, 2023, doi:10.3390/pharmaceutics15041192_

Round 1
Reviewer 1 Report
The review is well written and well explained the topic “Nanoparticle-mediated delivery approach of metronomic chemotherapy: Mechanism, challenges, recent advances, and future 3 perspectives”. I recommend its publication.
Author Response
Author response to reviewer comments
The authors are very thankful to the Editor, the Editorial team, and the Reviewers for consideration of our manuscript and for providing their valuable suggestions.
We have now addressed all the comments and incorporated the required changes into a revised version (highlighted in red font) as described in the point-by-point response below. These changes have helped to improve the manuscript substantially, which, we now hope is acceptable for publication in the Pharmaceutics journal.
Reviewer: 1
Comments to the Author
The review is well written and well explained the topic “Nanoparticle-mediated delivery approach of metronomic chemotherapy: Mechanism, challenges, recent advances, and future 3 perspectives”. I recommend its publication.
Author response: We are extremely grateful for reviewer 1 for finding our review well-written and well-explained. We also like to thank the reviewer for the recommendation of publication.

Reviewer 2 Report
Please add references and details in regard to the classifications and different types of nanoparticles. A recent review can be found at DOtl:10.1080/21691401.2019.1577885
There are a number of the use of enzymes as chemotherapeutic agents in nanoparticles that should ne included: see above reference for details
Author Response
Author response to reviewer comments
The authors are very thankful to the Editor, the Editorial team, and the Reviewers for consideration of our manuscript and for providing their valuable suggestions.
We have now addressed all the comments and incorporated the required changes into a revised version (highlighted in red font) as described in the point-by-point response below. These changes have helped to improve the manuscript substantially, which, we now hope is acceptable for publication in the Pharmaceutics journal.
Reviewer: 2
Comments to the Author
Please add references and details in regard to the classifications and different types of nanoparticles. A recent review can be found at DOl:10.1080/21691401.2019.1577885
There are a number of the use of enzymes as chemotherapeutic agents in nanoparticles that should ne included: see above reference for
Author response: We appreciate reviewer 2 suggestion to add reference (DOl:10.1080/21691401.2019.1577885 ) to cite the classification and different types of nanoparticles. Indeed, this reference suggestion is very relevant to our topic and content, and we have now cited in the revised manuscript. In the revised review, this reference is in the list at number 17.
We have also included few enzymes as chemotherapeutic agents in nanoparticles citing the aforementioned reference.

Reviewer 3 Report
Manuscript ID: pharmaceutics-2245080
Title: Nanoparticle-mediated delivery approach of metronomic chemotherapy: Mechanism, challenges, recent advances, and future perspectives
Author: Vijay Kumar Panthi, Kamal Dua, Sachin Singh, Gaurav Gupta, Philip M. Hansbro, Keshav Raj Paudel
Overview and general recommendation:
The review highlights the importance of metronomic chemotherapy over the conventional chemotherapy approach, its mechanism, challenges, recent advances, and future perspectives.
I am unconvinced about the relationship between nanoparticles and metronomic chemotherapy; the review didn’t display this correlation. So, from my point of view, the study should focus on metronomic chemotherapy, its mechanism, challenges, recent advances, and future perspectives without adding nanoparticles-mediated delivery. Otherwise, I found the review well-designed and written, except for typos, grammar mistakes, and punctuation. However, I ask the authors to address the recommendations for improving the manuscript before acceptance for publication.
Comments to the authors:
1) The title didn’t reflect the aim of the review.
2) The introduction section lacks details about the metronomic chemotherapy and nanoparticle-mediated delivery, which are the review's core.
3) On line 110, the author should mention the full name instead of the repeated abbreviation.
4) The numbering of the heading and subheadings in the manuscript needs revision.
5) On line 116, what do the authors mean by “they” in the sentence?
Author Response
Author response to reviewer comments
The authors are very thankful to the Editor, the Editorial team, and the Reviewers for consideration of our manuscript and for providing their valuable suggestions.
We have now addressed all the comments and incorporated the required changes into a revised version (highlighted in red font) as described in the point-by-point response below. These changes have helped to improve the manuscript substantially, which, we now hope is acceptable for publication in the Pharmaceutics journal.
Reviewer: 3
Manuscript ID: pharmaceutics-2245080
Title: Nanoparticle-mediated delivery approach of metronomic chemotherapy: Mechanism, challenges, recent advances, and future perspectives
Author: Vijay Kumar Panthi, Kamal Dua, Sachin Singh, Gaurav Gupta, Philip M. Hansbro, Keshav Raj Paudel
Overview and general recommendation:
The review highlights the importance of metronomic chemotherapy over the conventional chemotherapy approach, its mechanism, challenges, recent advances, and future perspectives.
I am unconvinced about the relationship between nanoparticles and metronomic chemotherapy; the review didn’t display this correlation. So, from my point of view, the study should focus on metronomic chemotherapy, its mechanism, challenges, recent advances, and future perspectives without adding nanoparticles-mediated delivery. Otherwise, I found the review well-designed and written, except for typos, grammar mistakes, and punctuation. However, I ask the authors to address the recommendations for improving the manuscript before acceptance for publication.
Author response: We are extremely grateful for reviewer 3 for the suggestion and comments below. We have now addressed all valid comments raised by the reviewer 3.
Comments to the authors:
- The title didn’t reflect the aim of the review.
Author response: We felt that the word “nanoparticles” is not correctly reflecting our aim of the review. We have changed the word “nanoparticles” to “nanoformulations” for wider coverage of the topic.
- The introduction section lacks details about the metronomic chemotherapy and nanoparticle-mediated delivery, which are the review's core.
Author response: We agree with reviewer point, and now we have added more information about the metronomic chemotherapy and nanoparticle mediated delivery in the introduction section highlighted with red font.
- On line 110, the author should mention the full name instead of the repeated abbreviation.
Author response: We apologies for this carelessness. We have now mentioned the full name of the abbreviation when they appear first time in the manuscript.
- The numbering of the heading and subheadings in the manuscript needs revision.
Author response: We have now revised the headings and subheading throughout the manuscript.
- On line 116, what do the authors mean by “they” in the sentence?
Author response: We are sorry for the ambiguous word “they”. “They” is referring to metronomic chemotherapy (MCT). We have now explained it for the clarity.

Reviewer 4 Report
I CONSIDER THAT THE TITLE OF THE PAPER DOES NOT ENCLOSE AND/OR REFLECTS THE BIBLIOGRAPHICAL REVIEW CARRIED OUT.
THE USE OF NANOARTICLES FOR ANTI-CANCER DRUGS IS MENTIONED IN A VERY SUPERFICIAL WAY AND VERY LITTLE DATA ARE PROVIDED ON SIZES, SIZE EFFECTS, CHARACTERIZATION, SYNTHESIS METHODS, ETC.
IF THE TERM OF NANOPARTICLES IS CONSIDERED IN THE TITLE, MORE SPECIFIC AND RELEVANT INFORMATION ON THE SUBJECT SHOULD BE PROVIDED
AN EXAGGERATED USE OF THE WORDS IS MADE: FUTHERMORE AND MOREOVER IN A GREAT PART OF THE ARTICLE WRITING, SYNONYMS MAY BE USED TO REPLACE WORDS THAT ARE REPEATED FREQUENTLY.
Author Response
Author response to reviewer comments
The authors are very thankful to the Editor, the Editorial team, and the Reviewers for consideration of our manuscript and for providing their valuable suggestions.
We have now addressed all the comments and incorporated the required changes into a revised version (highlighted in red font) as described in the point-by-point response below. These changes have helped to improve the manuscript substantially, which, we now hope is acceptable for publication in the Pharmaceutics journal.
Reviewer: 4
Comments to the Author
I CONSIDER THAT THE TITLE OF THE PAPER DOES NOT ENCLOSE AND/OR REFLECTS THE BIBLIOGRAPHICAL REVIEW CARRIED OUT.
Author response: We agree with reviewer comments, and this was also suggested by another reviewer. We felt that the word “nanoparticles” is not correctly reflecting our aim of the review. We have changed the word “nanoparticles” to “nanoformulations” for wider coverage of the topic.
THE USE OF NANOARTICLES FOR ANTI-CANCER DRUGS IS MENTIONED IN A VERY SUPERFICIAL WAY AND VERY LITTLE DATA ARE PROVIDED ON SIZES, SIZE EFFECTS, CHARACTERIZATION, SYNTHESIS METHODS, ETC.
Author response: In line with reviewer comments, we have now elaborated more on the use of nanoparticles for anti-cancer drugs including most of the recently published relevant articles.
IF THE TERM OF NANOPARTICLES IS CONSIDERED IN THE TITLE, MORE SPECIFIC AND RELEVANT INFORMATION ON THE SUBJECT SHOULD BE PROVIDED
Author response: We agree with reviewer comments. We have now expanded the relevant information on nanoparticles citing appropriate papers.
AN EXAGGERATED USE OF THE WORDS IS MADE: FUTHERMORE AND MOREOVER IN A GREAT PART OF THE ARTICLE WRITING, SYNONYMS MAY BE USED TO REPLACE WORDS THAT ARE REPEATED FREQUENTLY.
Author response: We have now proofread whole manuscript and tried our best to rectify the use of exaggerated words. We have also replaced a plenty of words with their synonyms highlighted with red color font.

Reviewer 5 Report
The study was well-designed, and the results are detailed enough to examine the mechanisms behind the reported trends. There is a need to justify the following comments, in order to understand their practicability for the stated objectives.
Comments to the Author
To improve the article’s quality, some editing of the English language is required throughout the manuscript.
The abstract must be rewritten to be representative of the whole paper (background and objectives, significance of study, limitations, main findings, future prospectives).
The abstract is too long. Please concise the abstract and only provide the main findings.
The phrase "are not effective to completely cure various cancers" could be improved to "have limited effectiveness in completely curing various cancers."
The phrase "these type of formulation in metronomic dose" should be "these types of formulations in metronomic doses."
Please provide a clear statement on the utility and novelty of this work in the abstract.
In introduction section the authors should clarify the economic importance in this study
The introduction should be enriched with recent references
The aims of the study are not clear in the introduction. Please further elaborate on the aims and give more details.
Avoid general information in the section “1.2 Conventional Chemotherapy and its limitation”.
Improve figure 1. Add one more point in “immune system”
Please provide the captions below table 1. MCT????
Please add the captions in table 2.
The novelty of the manuscript is questionable as already a lot of literature has been published. Please highlight the novelty of the manuscript
The conclusion needs to be improved. The conclusion is too general and short. What were the outcomes and prospects are not clear? The authors should justify elaborately how the current study is different from previous studies and useful to the scientific community.
Author Response
Author response to reviewer comments
The authors are very thankful to the Editor, the Editorial team, and the Reviewers for consideration of our manuscript and for providing their valuable suggestions.
We have now addressed all the comments and incorporated the required changes into a revised version (highlighted in red font) as described in the point-by-point response below. These changes have helped to improve the manuscript substantially, which, we now hope is acceptable for publication in the Pharmaceutics journal.
Reviewer: 5
The study was well-designed, and the results are detailed enough to examine the mechanisms behind the reported trends. There is a need to justify the following comments, in order to understand their practicability for the stated objectives.
Comments to the Author
To improve the article’s quality, some editing of the English language is required throughout the manuscript.
Author response: We have proofread the whole manuscript to rectify “typos” “cryptic text” “grammatical errors”. The changes are shown as red font.
The abstract must be rewritten to be representative of the whole paper (background and objectives, significance of study, limitations, main findings, future prospectives).
Author response: We have restructured the abstract according to the suggestion from reviewer 5. We believe that the flow of information in revised abstract looks better than previous version.
The abstract is too long. Please concise the abstract and only provide the main findings.
Author response: We agree with reviewer points. We have restructured and make the abstract more concise.
The phrase "are not effective to completely cure various cancers" could be improved to "have limited effectiveness in completely curing various cancers."
Author response: We are extremely grateful for reviewer suggestion, and we have replaced it as suggested. Abstract Line 21-22
The phrase "these type of formulation in metronomic dose" should be "these types of formulations in metronomic doses."
Author response: We are extremely grateful for reviewer suggestion. We have removed it and re-structured the abstract as per the reviewer suggestion.
Please provide a clear statement on the utility and novelty of this work in the abstract.
Author response: We have added a sentence to highlight the utility and novelty of our work in the abstract.
In introduction section the authors should clarify the economic importance in this study
Author response: This is a very important point, and we agree with reviewer suggestion. We have now added a paragraph highlighting the “economic importance of our study” in the introduction section.
The introduction should be enriched with recent references
Author response: We have now enriched the most recent reference not only on only introduction but also on other section throughput the manuscript wherever relevant.
The aims of the study are not clear in the introduction. Please further elaborate on the aims and give more details.
Author response: The aims the study is now highlighted in abstract and later in the introduction section.
Avoid general information in the section “1.2 Conventional Chemotherapy and its limitation”.
Author response: The already well know and very general information in the section “1.2 Conventional Chemotherapy and its limitation” has been deleted.
Improve figure 1. Add one more point in “immune system”
Author response: We have added more point “Inhibiting tumor-stromal activation” in “immune system”
Please provide the captions below table 1. MCT????
Author response: We have now added the captions below the table 1. MCT stands for metronomic chemotherapy and we have now elaborated it.
Please add the captions in table 2.
Author response: We have now added the captions below the table 2
The novelty of the manuscript is questionable as already a lot of literature has been published. Please highlight the novelty of the manuscript
Author response: We have highlighted the novelty of the manuscript in the abstract.
The conclusion needs to be improved. The conclusion is too general and short. What were the outcomes and prospects are not clear? The authors should justify elaborately how the current study is different from previous studies and useful to the scientific community.
Author response: We are re-written the conclusion section. The final suggestion from author point of view and future prospects are more elaborated. We have also justified how our works is different from previous studies and useful to the scientific audience.

Round 2
Reviewer 3 Report
The authors addressed all the raised comments. So, I recommend the acceptance of the manuscript for publication.
Reviewer 4 Report
the observations were attended
Reviewer 5 Report
Accepted